# The Role of Phosphatidylinositol Phosphate Kinases during Viral Infection

**DOI:** 10.3390/v12101124

**Published:** 2020-10-03

**Authors:** Anne Beziau, Denys Brand, Eric Piver

**Affiliations:** 1INSERM U1259, University of Tours, 37000 Tours, France; anne.beziau@etu.univ-tours.fr (A.B.); denys.brand@univ-tours.fr (D.B.); 2Virology Laboratory, Tours University Hospital, 3700 Tours, France; 3Biochemistry and Molecular Biology, Tours University Hospital, 3700 Tours, France

**Keywords:** phosphatidylinositol (PI), phosphatidylinositol phosphate kinases (PIPK), phosphatases, viral replication

## Abstract

Phosphoinositides account for only a small proportion of cellular phospholipids, but have long been known to play an important role in diverse cellular processes, such as cell signaling, the establishment of organelle identity, and the regulation of cytoskeleton and membrane dynamics. As expected, given their pleiotropic regulatory functions, they have key functions in viral replication. The spatial restriction and steady-state levels of each phosphoinositide depend primarily on the concerted action of specific phosphoinositide kinases and phosphatases. This review focuses on a number of remarkable examples of viral strategies involving phosphoinositide kinases to ensure effective viral replication.

## 1. Introduction

Phosphoinositides (PPIns) play a key role in cell physiology, even though they account for only 10% of cellular phospholipids [1]. They act as second messengers and are involved in several cell-signaling pathways, actin cytoskeletal remodeling and membrane dynamics. Intracellular PPIns levels are regulated by phosphatases and kinases, to ensure the correct functioning of cellular processes. The distribution of phosphatidylinositol phosphate kinases (PIPK) and PIP phosphatases between the various subcellular compartments influences the location of PPIns and explains the extensive involvement of PPIns in numerous metabolic pathways within cells [2,3]. Viruses are obligate parasites of cells, in which they find the elements they require for replication. The key roles of PPIns within the cell may account for the close interaction between these molecules and viral proteins or viral replication steps. Furthermore, lipid metabolism and transport, and lipid-mediated signal transduction are disrupted by viral replication. Many viruses are known to need cellular lipids to complete their life cycles. The disruption of lipid pathways, diverting cellular lipid metabolism to the advantage of the virus, may favor viral replication and assembly, and enable viruses to evade the host immune system to become more infectious.

PPIns are present during different stages of the viral cycle occurring in different parts of the cell. It is, therefore, crucial to determine the importance of kinases involved in lipid metabolism in the cycles of different viruses.

## 2. Phosphatidylinositols (PI)

Phosphatidylinositols (PI) are the precursors of PPIns, phosphorylated derivatives of PI known to play an important role in the life cycle of the cell. PPIns are tightly regulated, temporally and spatially, by a set of kinases and phosphatases (Figure 1) distributed between the various cell compartments. The subcellular distribution of PPIns determines the roles of these molecules in different metabolic pathways (Figure 2).

### 2.1. Metabolism

PIs are metabolized into seven different PPIns. PIs are synthesized in the endoplasmic reticulum (ER) and are then subsequently transported throughout the cell. They are synthesized by PI synthase (PIS), from diacylglycerol and free inositol, through the formation of a phosphodiester bond involving the -OH group in the D1 position of the ring [4,5]. Despite the presence of five free hydroxyl groups on the inositol nucleus, phosphorylation occurs only at positions -3, -4, and -5 (-OH groups) of the ring. PIs account for 10–20% of cellular phospholipids in the eukaryotic cell, whereas PPIns account for only 1% of all cellular phospholipid species [1,5,6].

The most abundant of the seven PPIns in cells is PI(4,5)P_2_. It can be produced in three different ways: by dephosphorylation at position D3 by PIP3-phosphatase, or by phosphorylation at position D4 of PI(5)P by phosphatidylinositol-5-phosphate-4-kinase (PI5P4-kinase also called PIP5KII), but mostly by phosphorylation at position D5 of PI(4)P by phosphatidylinositol-4-phosphate-5-kinase I (PIP5KI) (Figure 1) [1]. The PIP5KI family of kinases contains three isoforms (α, β, γ) widely distributed in cells and subject to feedback regulation by many factors [7].

All PIs are involved in biological processes. Their location within the cell determines their interactions with different effectors and, thus, their roles. One important pathway of PI(4,5)P_2_ metabolism involves the cleavage of this molecule by phospholipase C (PLC) to generate diacylglycerol (DAG) and inositol 3 phosphate (IP3), which are involved in the release of Ca^2+^ linked to an increase in virion release [8,9].

### 2.2. Localization and Functions

The distribution and composition of PPIns differs between cell types, production sites and enzyme locations. As a result, different PPIns have different subcellular distributions (Figure 2) [10]. For example, PI(4)P is associated with the Golgi apparatus, the trans-Golgi network (TGN) involved in vesicular transport, and it is also found in the inner leaflet of the plasma membrane (PM) [11]. PI(5)P has also been implicated in endosomal trafficking [6]. PI(3)P, another important mono-phosphorylated PPIn, is involved in endosomal motility [12].

There are three types of biphosphorylated PPIns: PI(4,5)P_2_, PI(3,5)P_2_ and PI(3,4)P_2_. Within cells, PI(4,5)P_2_ is the most abundant biphosphorylated PPIn, the other two types being barely detectable. PI(3,4)P_2_ is found at the PM or endosome, and is involved in actin remodeling. PI(3,5)P_2_, which is located in the endosome membrane, is involved in actin cytoskeleton remodeling and protein transport to late endosomes [13]. PI(4,5)P_2_, which accounts for about 45% of total PPIns, is found mostly at the PM, with a gradient in the abundance of this molecule from the nucleus to the internal leaflet of the PM. It is also involved in endosomal trafficking, actin cytoskeleton remodeling and cell migration [14]. Finally, PI(3,4,5)P_3_, the only triphosphorylated PPIn, is located at the PM and involved in cell survival and cellular motility [15].

The functions and distribution of PPIns depend on the kinases that regulate them.

## 3. Phosphatidylinositol Kinase

The production and regulation of all PPIns depend on several families of phosphatidylinositol kinases (PIKs) and phosphatidylinositol phosphatases (PIPs). These various families of kinases have specific substrates and distributions, ensuring the specific production of PPIs. In humans, there are 19 genes encoding PIKs, including a number of isoforms, which also have specific features [16]. Regulation of the activation and localization of PIK isoforms is therefore crucial for the accurate synthesis of PPIns required for different cellular signaling pathways.

### 3.1. PI3K Family

The PI(3)-kinases (PI3K), are responsible for phosphorylation at the D3 position of the inositol ring and can be classified into three types on the basis of phospholipid substrate affinity. The PI3K type I (also named PI3KC) family has four members, the PI3KCα, β, δ and γ isoforms, which mostly phosphorylate PI(4,5)P_2_ to generate PI(3,4,5)P_2_ [17]. PI3K enzymes have several regulatory subunits, which stabilize the catalytic subunits and regulate their activities [18]. The regulatory subunit p85, for example, has been shown to mediate differential regulation of the activities of the PI3KCα, β and δ isoforms [19]. These isoforms can also be activated differently by membrane signaling proteins, such as receptors with tyrosine kinase activity or GTPases of the Ras family [20]. The involvement of PI(3,4,5)P_3_ in diverse pathways can thus be explained, in part, by the activation and control of signaling channels. Moreover, the identification of activating mutations of the *pi3kc*-α gene in patients developing various tumors, and the activation of PI3Ks by different oncogenes demonstrate that increases in PI3K signaling are one of the hallmarks of cancer occurrence [21,22].

The PI3K type II family contains three isoforms (PI3KCIIα, β and γ that generate PI(3)P and PI(3,4)P_2_ by phosphorylating PI and the PI(4)P in position D3 of the inositol ring) [23]. The isoforms of PI3KCII are involved in insulin secretion, cell survival and proliferation, and in the endocytosis of clathrin-coated vesicles [24,25,26,27]. The PI(3,4)P_2_ produced by the PI3KCIIα isoform is essential for the maturation of clathrin-coated pits. In this process, PI(3,4)P_2_ is produced to recruit the “sorting nexin 9” proteins that act, with dynamin, in the processing and cleavage of clathrin from endocytic vesicles [28].

Type III PI3K, known as PI3KCIII (Vps34), specifically catalyze the phosphorylation of PIs to PI(3)P [29]. This family of kinases can form two different protein complexes, regulating the activity of the enzyme in different pathways [30]. Complex 1 (composed of PI3KC3, p150, Beclin1, autophagy-related 14 -like protein (Atg14L) and nuclear receptor-binding factor 2 (NRBF2)) controls the formation of autophagosomes, whereas complex 2 (composed of PI3KC3, p150, Beclin1 and UV radiation resistance-associated gene protein (UVRAG)) is involved in retrograde vesicular transport [31,32].

### 3.2. PI4K Family

Phosphorylation at position D4 of the inositol ring is performed by the PI(4)-kinases (PI4K). The PI4K type III family is composed of two isoforms, PI4K IIIα and β which participate in the formation of PI(4)P at the plasma membrane, or in the Golgi apparatus and TGN, respectively [33,34]. The activity of PI4K type III, like that of other PI kinases, is regulated by cellular partners. One particular feature of PI4K enzymes is that they are regularly hijacked by positive-strand RNA viruses for the generation of organelles enriched in PI4P [35]. PI(4)P can also be generated by a second family of PI kinases, PI4K type II (PI4K II). This family consists of two isoforms, α and β, which are found mostly in the intracellular membranes. PI4KIIα is the most active isoform, responsible for the production of almost half of all the PI(4)P generated [36]. The alpha isoform, which is located principally in the Golgi apparatus and endosomes, is involved in endosome trafficking. Endocytic trafficking requires membrane fusion between the vesicles and the subcellular target compartments. This fusion is ensured by SNARE (soluble *N*-ethylmaleimide-sensitive factor attachment protein receptor), which is present on the vesicles and membranes of the target compartments. Fusion efficiency is also dependent on the lipid composition of the membranes. The production of PI(4)P by PI4KIIα makes it possible to modulate the functions of the SNARE proteins and, thus, to control the trafficking of endosomal structures [37,38,39]. PI4KIIβ has been less studied, but has nevertheless been implicated in the activation of early T cells via the CD3-TCR (T Cell Receptor) signaling pathway [40].

### 3.3. PIP5KI Family

PI(4,5)P_2_ is involved in a very broad array of key biological functions. It is present mostly in the inner leaflet of the plasma membrane, but numerous clusters of specifically distributed PI(4,5)P_2_ have been identified, suggesting that its production may be subject to spatiotemporal regulation. It is produced essentially by the phosphorylation of PI(4)P by kinases of the PIP5K type 1 family (PIP5KI) [41,42]. PIP5KI enzymes are encoded by three genes, *pip5kα*, *β* and *γ*. The *pip5kα* gene generates six splice variants. The human PIP5Kα isoform corresponds to the PIP5KIα isoform in mice, and vice versa. The nomenclature adopted here is that used in humans. The central region of PIP5KI, which contains the catalytic domain, is highly conserved in all three isoforms, whereas variations of the N-terminal and C-terminal regions of the protein determine its subcellular distribution, to ensure the localized production of PI(4,5)P_2_ [43]. The activity and individual locations of the PIP5KI isoforms are also regulated by different membrane receptors, GTPases of the Rho, Rac and Arf families, or by phosphorylation processes [44].

#### 3.3.1. PIP5KI Alpha (PI5PKIα)

The PIP5KIα isoform has a molecular weight of about 68 kDa. Three alternative splicing variants have been identified, the roles of which have yet to be clearly characterized [45]. The PIP5KIα isoform is usually located on the inner leaflet of the PM and at the Golgi apparatus. Several other cellular factors specifically regulate the enzymatic activity and localization of PIP5KIα. For example, the stimulation of cellular G protein-coupled receptors leads to the translocation of this isoform across the membrane and an increase in its local activity [46]. The activity of PIP5KIα at the plasma membrane, thus, regulates numerous processes by ensuring the localized production of PI(4,5)P_2_. This enzyme is also involved in the mechanism of phagocytosis, in which it regulates the dynamics of the actin cytoskeleton to allow phagosome formation [47,48]. The formation of clathrin-coated endocytotic vesicles is also regulated by the α isoform of PIP5KI [49]. The remodeling of the actin cytoskeleton provided by PIP5KIα and coordinated by Rac1 is essential for budding [50]. PIP5KIα is also located in the “nuclear speckles” observed during the processing of certain mRNAs [51,52,53].

#### 3.3.2. PIP5KI Beta (PIP5KIβ)

PIP5KIβ is found at the plasma membrane but seems to be localized mostly in vesicular structures [43]. The PIP5KIβ isoform has a molecular weight of approximately 68 kDa, similar to that of PIP5KIα.

A recent study showed that the inhibition of PI(4,5)P_2_ production by a knockout of PIP5KIβ could be compensated by PIP5KIα expression [2]. These results, again, highlight the cooperation between the various PIP5KI isoforms in signaling pathways. Inhibition of the beta isoform is compensated by PIP5KIα [2]. Finally, PIP5Kiα β heterodimers have been observed in vitro, and the structure of the dimerization interface of PIP5KIα has been characterized [2,3]. The formation of these heterodimers modulates the activity and localization of PIP5KIβ, suggesting that cooperation occurs between the different isoforms of PIP5KI [3].

#### 3.3.3. PIP5KI Gamma (PIP5KIγ)

PIP5KIγ is the most diverse isoform of PIP5KI. Six human splice variants (PIP5KI-v1 to v6) have been identified, which differ in terms of the length of their C-terminal ends [1,54,55]. The PIP5KIγ-v1 protein has a molecular weight of 87 kDa (PIP5KI 87) and is the major form in most tissues. The 90-kDa PIP5KI-v2 protein (PIP5KI 90) is the second predominant form of the protein in most tissues. PIP5KIγ-v3 (93 kDa) is specifically localized in neuronal cells from various regions of the brain [54]. The PIP5KIγ 87, PIP5KIγ 90 and PIP5KIγ-v3 isoforms are all located at the PM. The distribution of the variants of PIP5Kγ is consistent with the location of PIP5KIα and β. The interaction of PIP5KIγ 90 with the Four-band-Ezerin-Radixin-Moesin domain (named FERM domain) of talin ensures its site-specific focal adhesion and its enzymatic activation. PI(4,5)P_2_ interacts locally with actin-associated proteins, activating these proteins. For example, it mediates the binding of talin to PM and its interaction with integrin at focal adhesion sites [56,57]. Conversely, the overexpression of PIP5KIγ 90 leads to the disassembly of focal adhesions [58]. The dynamics of structural assembly and disassembly therefore require tight control over the PIP5KIγ 90-talin interaction. PI(4,5)P_2_ is required for the formation of clathrin-coated vesicles during endocytosis, due to its roles in actin remodeling and the clustering of the clathrin adaptor protein 2 (AP2). The observation of a direct interaction between the γ isoform and the AP2 complex suggests that PIP5KIγ is specifically recruited for PI(4,5)P_2_ production at the endocytosis bud [59]. In addition, the isoform γ, like PIP5KIα, is also involved in calcium-dependent exocytosis. Indeed, the influx of Ca^2+^ leads to the recruitment and activation of PIP5Kγ by ADP-ribosylation factor 6 (Arf6), triggering the production of PI(4,5)P_2_ required for this process [60]. Similarly, PIP5KIγ 87 controls Ca^2+^ flux in HeLa cells, through the compartmentalized production of PI(4,5)P_2_, which is used as a precursor for the production of PI(1,4,5)P_3_ [61]. Finally, PIP5KIγ has been shown to play a key role in the formation of adherens junctions, which are essential for the polarization of epithelial cells. PIP5KIγ can interact directly and simultaneously with E-cadherin and proteins of the AP complex, to ensure the local production of PI(4,5)P_2_ necessary for junction formation and regulation of the basolateral trafficking of E-cadherin [14]. The deregulation of PIP5Kγ expression and, thus, of cell-cell junctions, is associated with the development of breast cancers, in particular [62].

### 3.4. PIKfyve

Finally, the last major type of PI kinase comprises the enzymes catalyzing the production of PI(4,5)P_2_ and PI(3,5)P_2_. Phosphatidylinositol 3-phosphate 5 kinase (PIKfyve) was first identified as responsible for the production of PI(5)P and PI(3,5)P_2_ in vitro [63]. However, it remains a matter of debate whether PI(5)P is produced directly under the action of the PIKfyve. An in vivo study showed that PIKfyve played an indirect role in PI(5)P synthesis, by producing all of the cellular PI(3,5)P_2_, which then served as a substrate for PI phosphatases [64]. PIKfyve ensures the phosphorylation of PI(3)P, by binding to its FYVE domain. Due to its ability to produce PI(3,5)P_2_, this enzyme is principally involved in trafficking, such as the transport of early endosomes to the TGN or the maturation of endosomes and their transport to lysosomes [65,66,67]. The production of PI(3,5)P_2_ leads to the recruitment of effector proteins, such as Rab9 p40 effectors or “sorting nexin-1” proteins, which are involved in the retrograde trafficking of late endosomes [65,66,67].

The involvement of these enzymes in the homeostasis of the cellular membrane and other crucial cellular pathways has led to viruses making use of them for their own multiplication cycle.

Here, we will describe the links between different viruses and the phosphoinositide metabolism necessary for their entry into cells and assembly. As mentioned above, PPIn metabolism is mediated by PI kinases, the key prerequisite for local enrichment in PPIns, favoring viral replication. Below, we will use examples to describe the links between PI kinases and the viral replication cycle.

## 4. Kinases and Viruses

Viruses are obligate parasites of the cell, which supplies them with all the elements necessary for their replication. Lipid metabolism, an essential metabolic pathway of cells, is disrupted by viral replication. Viruses take advantage of host lipid metabolism, reorganizing membrane lipid composition for their own benefit. We have therefore chosen virus examples for which replication or assembly sites have a close relationship with membrane domains where major types of PI kinases are located.

### 4.1. Coronavirus

Coronavirus particles are surrounded by a viral envelope acquired during budding and composed of structural proteins and cellular lipids. These viruses are known to interact closely with PI kinases, particularly at early stages of infection. SARS-CoV (1 and 2) and MERS-CoV are the most studied members of this family of viruses. PI4KIIIβ is known to be involved in the entry into cells of pseudoviruses bearing the SARS-CoV-1 spike protein (Table 1) [68]. Cellular entry, mediated by the ACE2 receptor (angiotensin I-converting enzyme 2), is strongly inhibited by the knockdown of PI4KIIIβ, suggesting an important role for this enzyme in the entry process. PI4KIIIβ acts not by binding the virus to the receptor, but by altering the composition of the lipid membrane of organelles. These alterations are crucial for the second part of the entry step. The resulting increase in PI(4)P levels within membrane organelles creates a lipid environment favorable for SARS-CoV-1 entry [68].

Two signaling pathways are involved in MERS-CoV replication. The extracellular signal-regulated kinase/mitogen-activated protein kinase (ERK/MAPK) and phosphoinositol 3-kinase/serine-threonine kinase/mammalian target of rapamycin (PI3K/AKT/mTOR) signaling networks play important roles in infection with this virus. These signaling networks are also known to be important for several cell regulatory responses, such as cell proliferation or apoptosis [69], and are targeted by a broad range of viral pathogens. Kindrachuk et al., demonstrated that various inhibitors of ERK/MAPK and PI3K/AKT/mTOR strongly inhibit MERS-CoV replication (Table 1) [70].

The SARS-CoV-2 pandemic has provided opportunities to discovery new links between coronaviruses and PI kinases. The entry mechanism of SARS-CoV-1 and 2 is dependent on the late endosomal compartment, which plays an essential role in effective infection. The two-pore channel 2 (TPC2), a specific marker of late endosomes, is important for SARS-CoV-2 entry [71], and SARS-CoV-2 infection is impeded by the inhibition of endosomal acidification. As described above, PIKfyve is involved in endosome maturation, and its inhibition with chemical inhibitors (apilimod) should impede SARS-CoV-2 infection (Table 1) [72]. Coronaviruses are not the only viruses to make use of the endosome pathway during their cycle. Ebola virus also uses this pathway, for replication.

### 4.2. Ebola Virus

Ebola viruses (EBOV) belong to the filovirus family. Like other members of this group, they have a filamentous morphology and a host-derived lipid envelope acquired during budding. EBOV enters the cell through interactions with late endosomes. Indeed, the viral glycoprotein of EBOV interacts with the cellular protein Niemann–Pick C1 (NPC1), which is located in late endosomes [73]. PIKfyve, through its role in endosome maturation, is essential for EBOV entry into the cells, as it mediates the transport of EBOV particles to NPC1-positive late endosomes. Indeed, without endosome maturation and fusion with lysosomes, the entry of EBOV into cells is blocked. Nelson et al. demonstrated the importance of PIKfyve in the EBOV virus replication cycle, using apilimod, an inhibitor of PIKfyve [74]. Apilimod blocks fusion with late endosomes, resulting in the accumulation of viral particles in early endosomes [73].

PI3K is another important kinase for Zaire Ebola virus (ZEBOV) entry into cells. Saeed et al. have reported this enzyme to play an important role at or during viral entry into the cell. The Akt pathway is involved in several cellular pathways and in the indirect regulation of PI3K [75]. The inhibition of PI3K or AKT significantly decreases ZEBOV entry, suggesting that the PI3K/AKT pathway plays an essential role in this process. Rac1, an actin polymerization regulator involved in endocytosis, also seems to be involved in ZEBOV entry [76].

The Viral Protein 40 (VP40) matrix protein of EBOV plays a key role in budding. VP40 is composed of two domains: an N-terminal domain involved in dimerization and oligomerization, and a C-terminal membrane-binding domain mediating binding to the PM through electrostatic interactions. Efficient interactions of VP40 with the PM require high concentrations of PI(4,5)P_2_, PI(3)P and PI(3,4,5)P_3_ (Table 1). Moreover, enrichment of the PM in PI(4,5)P_2_ induces membrane curvature through changes to lipid composition, leading to the assembly and budding of EBOV [77,78].

### 4.3. Hepatitis C Virus (HCV)

PPIns and kinases are also known to be involved in HCV replication. PI(4)P has been identified as a clinical hallmark of HCV infection. Liver biopsies on HCV patients have also revealed the presence of high levels of PI(4)P and a disruption of cellular distribution of PI(4)P [79]. In uninfected cells, PI4KIIIα generates a pool of PI(4)Ps mostly localized on ER membranes; by contrast, in infected cells PI(4)Ps are redistributed, forming a punctate distribution in the cytoplasm suggesting an important role for PI4KIIIα in HCV replication [80,81,82,83]. Non-structural protein 5A (NS5A), a replication complex protein, is directly involved in the increase in PI(4)P levels, by enhancing PI4KIIIα activity through direct interaction [68,80]. These studies indicate that NS5A recruits PI4KIIIα to the membranous replication compartment and stimulates PI4KIIIα activity, resulting in the robust induction of PI(4)P pools, which are required to maintain the integrity of the membranous web structure [80]. Berger et al. showed that the viral NS5A protein both recruits and activates PI4KIIIα, and that this activation is required for membranous web integrity, and viral replication [84]. Indeed, PI4KIIIα is an important host factor for the formation of membranous tissues enriched in PI(4)P with the physicochemical properties required for correct replication complex assembly [7]. Despite the considerable increase in cellular PI(4)P levels induced by HCV infection, the overall level of PI4KIIIα is not significantly altered, suggesting that the increase in PI(4)P levels results from an increase in the kinase activity of PI4KIIIα, rather than its abundance [85]. Indeed, PI4KIIIα activity is essential for HCV replication (Table 1).

PI(4)P is also indirectly involved in HCV secretion. Golgi phosphoprotein 3 (GOLPH 3) binds PI(4)P and an unconventional myosin (MYO18A) in the TGN. GOLPH3 and MYO18A mediate the attachment of the TGN to the actin cytoskeleton and participate in the efficient budding of vesicles from the TGN [80,86]. The inhibition of TGN vesicle production impairs HCV secretion. These data suggest that PI(4)P, and, thus, PI4KIIIα activity, operate at different steps of the HCV cycle. PI(4)P pools are recruited to maintain the integrity of the membranous web structure during HCV replication. The integrity of TGN vesicle trafficking, which is ensured by the presence of PI(4)P at the TGN membrane, is crucial for HCV secretion.

However, PI3K, which catalyzes the production of PI(3,4)P_2_ and PI(3,4,5)P_3_, is also important for the life cycle of HCV. HCV NS5A interacts with the PI3K regulatory subunit p85, releasing the inhibition of the catalytic p110 subunit, leading to the formation of PI(3,4)P_2_ and PI(3,4,5)P_3_ and allowing Akt recruitment. Akt, which is involved in cell survival and apoptosis inhibition, drives hepatocellular carcinoma development via the PI3K-Akt pathway [80].

### 4.4. Enterovirus

As for some positive-strand RNA viruses, the replication of enteroviruses is closely linked to cellular lipid metabolism. The replication of enterovirus, like that of HCV, requires an increase in the amount of cellular PI(4)P, a key component of the replication mechanism, mediated by PI4KIIIβ activity. Picornavirus (PV) is an enterovirus that has been shown to induce a significant increase in cellular phospholipid synthesis [35] and incorporation into the membrane, associated with a large decrease in cellular protein synthesis [87]. Some viral proteins, such as 3A, 3AB, 3CD and 3Dpol, have been shown to interact with PI(4)P. During PV infection and viral replication, PI4KIIIβ accumulates at the replication membrane. This leads to the formation of PI(4)P lipid-enriched organelles, which facilitate viral replication (Table 1). It has been suggested that this recruitment is dependent on the viral protein 3A, guanine nucleotide exchange factor 1 (GBF1) and ADP-ribosylation factor 1 (Arf1) activation by GBF1 [35,82]. The 3A viral protein coprecipitates with PI4KIIIβ [82], suggesting that these two proteins form a complex during enterovirus replication. Expression of the 3A viral protein enhances the membrane recruitment of Arf1 and PI4KIIIβ, inducing disassembly of the Golgi apparatus [82], and thus mimicking specific aspects of the viral infection phenotype. Another viral protein, 3Dpol, which is involved in enteroviral RNA replication, interacts specifically and with PI(4)P, with an affinity higher than that of other phospholipids.

Enteroviruses, and picornavirus in particular, have two major effects on cellular PI(4)P during infection. First, they increase the incorporation of the precursor of this molecule into the membrane, thereby enhancing PI [4] P synthesis by PI4KIIIβ. Second, by inducing the production of large amounts of PI(4)P, they cause the specific production of PI(4)P lipid-enriched organelles involved in viral replication (Table 1) [87].

### 4.5. Human Immunodeficiency Virus-1 (HIV-1)

Phosphatidylinositol kinases are known to be involved in HIV-1 entry. During this process, a fusion pore forms through the clustering of the CD4 receptor and the C-X-C Chemokine Receptor 4 (CXCR4) and C-C Chemokine Receptor 5 (CCR5) coreceptors, in an HIV-1-dependent mechanism. This clustering of CD4 and CXCR4 induces PIP5KIα activation, leading to the production of PI(4,5)P_2_, which modulates the actin cytoskeleton to promote HIV-1 entry (Table 1) [88]. The static actin cytoskeleton in resting T cells represents a barrier to HIV-1 infection and must be reorganized dynamically, by activating cofilin, an actin depolymerization factor, in particular, to promote virus infection [89]. PI(4,5)P_2_ can also interact with the proteins regulating the actin cytoskeleton, such as vinculin, α-actinin, talin and actin-capping proteins, suggesting a possible role for PI(4,5)P_2_ in regulating cytoskeletal dynamics [45]. PI(4,5)P_2_ inhibits proteins mediating the depolymerization or separation of filaments, such as gelsolin or cofilin. However, it also triggers the activation of signaling pathways, leading to filament polymerization [90]. Given the role of actin in endocytosis, PI(4,5)P_2_ and, thus, the isoforms of PIP5KI play a critical role in the entry of the virus into the cell. In addition to activating PIP5KIα through clustering, the gp120 glycoprotein induces PI(4,5)P_2_ production by PIP5KIα during viral contact with the cell [91]. Through its role in actin cytoskeleton regulation, PI(4,5)P_2_ also contributes to the completion of HIV-1 reverse transcription and assembly of the pre-integration complex [88]. Another important step in the HIV-1 life cycle involving PPIns is the assembly of the virus from the Gag polyprotein precursor (Pr55Gag). This precursor is a polyprotein with several protein domains: a matrix (MA or p17) protein displaying *N*-terminal myristoylation for PM interaction, and a highly basic region (HBR), spanning residues 17–31 [92,93,94,95,96], for interaction with PPIns, including PI(4,5)P_2_ [95,97,98] in particular, a capsid (CA or p24) protein responsible for Gag oligomerization (Gag-Gag interaction) mediating the formation of virus-like particles (VLPs), a nucleocapsid (NC or p7) involved in the encapsidation of unspliced genomic RNA (gRNA), two smaller spacer peptides, SP1 and SP2, and p6, which is involved in the budding of viral particles via the cellular endosomal secretory complexes required for transport (ESCRT) complex. During the assembly of viral particles, Pr55Gag interacts with regions of the inner leaflet of the plasma membrane composed of certain lipids, including PI(4,5)P_2_ (Table 1).

Assembly and budding occur at the PM by the same mechanisms in two major cell types: CD4^+^ T cells and macrophages. Assembly occurs at the PM in both these cell types, but at different sites. Indeed, in T cells, as in HeLa cells, assembly takes place at the PM itself, whereas, in macrophages, assembly occurs in the intracellular plasma membrane-connected (IPMC) compartment, which is also known as the virus-containing compartment (VCC). This particular location for virus assembly undoubtedly enables HIV-1 to escape the immune system. HIV-1 assembly occurs at the PM, in microdomains known as lipid rafts, composed of specific lipids: PI(4,5)P_2_, cholesterol, glycosphingolipids and phospholipids [88,98,99]. These microdomains also contain a tetraspanin-enriched domain (TEM) composed essentially of CD81 [88]. Thus, the lipid composition of HIV-1 is very similar to that of lipid rafts [100]. Pr55Gag seems to mediate association with the lipid raft, and is sufficient for the assembly and release of viral pseudoparticles [101]. The budding of HIV-1 therefore involves a specific process of raft grouping [100]. Pr55Gag is produced in a compact form in the cytosol, and gRNA is linked to NC and the HBR domain, inducing regulation by binding inhibition in the absence of PI(4,5)P_2_ [97,102,103]. The HBR-PI(4,5)P_2_ interaction at the PM induces a switch to the extended form of Gag, causing MA myristate to be exposed at the plasma membrane and stabilizing the interaction between Gag and the PM [98]. PI(4,5)P_2_ seems to stabilize Gag-MP interaction through conformational changes, the enhancement of multimerization or both [104,105].

As previously described, PI(4,5)P_2_ is mostly produced by enzymes of the PIP5KI family, which contains three isoforms (PIP5KIα, PIP5KIβ and PIP5KIγ), although PIP5KIα is the principal isoform produced (Table 1). Using siRNAs directed against each of the PIP5KI isoforms, Gonzales et al. showed that the depletion of PIP5KIα in HeLa cells induces the relocation of Pr55Gag from the PM for degradation by the proteasome, lysosome pathways or both [106]. Consistent with this observation, the overexpression of polyphosphoinositide 5-phosphatase IV (5PtaseIV), which dephosphorylates polyphosphoinositide, induces a depletion of PI(4,5)P_2_, inhibiting virus production [107]. Furthermore, when PIP5KIγ is inhibited, Pr55Gag is not degraded, but is instead relocated to intracellular vesicles corresponding to late endosomes [106]. All these data, therefore, demonstrate the importance of PI(4,5)P_2_ and the PI5KIα and γ isoforms for the HIV-1 cycle, particularly for the entry phase, through their effects on the actin cytoskeleton, and during the early steps of virus assembly, in which they stabilize Pr55Gag at the PM. These results confirm the involvement of PI(4,5)P_2_ and PI kinase in diverse cellular functions, including vesicular trafficking, and in PM structures linked to the cytoskeleton.

## 5. Conclusions

Studies concerning the interaction between PPIns metabolism and the viral multiplication cycle are improving our understanding of how viruses make use of the metabolic pathways of the cells they infect.

PPIns metabolism plays an important role in the life cycles of several positive-strand RNA viruses, but each virus makes use of different PI kinases and PPIns. PIKfyve is required for effective EBOV and coronavirus infections, and PI3K is required for EBOV replication. PI4K is involved in the replication of HCV and enterovirus, through PI(4)P-enriched organelles, with its membrane remodeling effects facilitating effective coronavirus entry. Regarding HCV and enterovirus, the increase in PI4K activity creates a localized pool of PPIns favoring their replication or their assembly.

PIP5KI is also involved in HIV-1 entry, through actin cytoskeleton remodeling, and in assembly at the PM. HIV-I assembly occurs at the PM, but at different sites in T cells and macrophages. The factors governing this phenomenon include differences in the cellular distribution of PIP5KI isoforms and, thus, of PI(4,5)P_2_ between CD4^+^ T cells and macrophages. Further studies of the isoforms of PIP5KI and the metabolism of PI(4,5)P_2_ are required to test confirm this hypothesis. The viral envelope is derived from the PM of the infected cell and the composition of the PM differs between cell types. The lipid composition of the viral envelope therefore differs between cell types. These differences may explain the difference in the location of HIV-1 assembly between T cells and macrophages [101], and the differences in the distribution of PPIns. Some of the effects of PIP5KI on HIV-1 assembly have been deciphered in T cells, but the role of PIP5KI in HIV-1 assembly in macrophages requires further characterization. Taken together, these results suggest that these PI kinases are potentially useful therapeutic targets for blocking viral replication. However, they are also crucial elements for the correct functioning of the cell, so caution will be required in the development of therapeutic approaches targeting them.

## Figures and Tables

**Figure 1 viruses-12-01124-f001:**
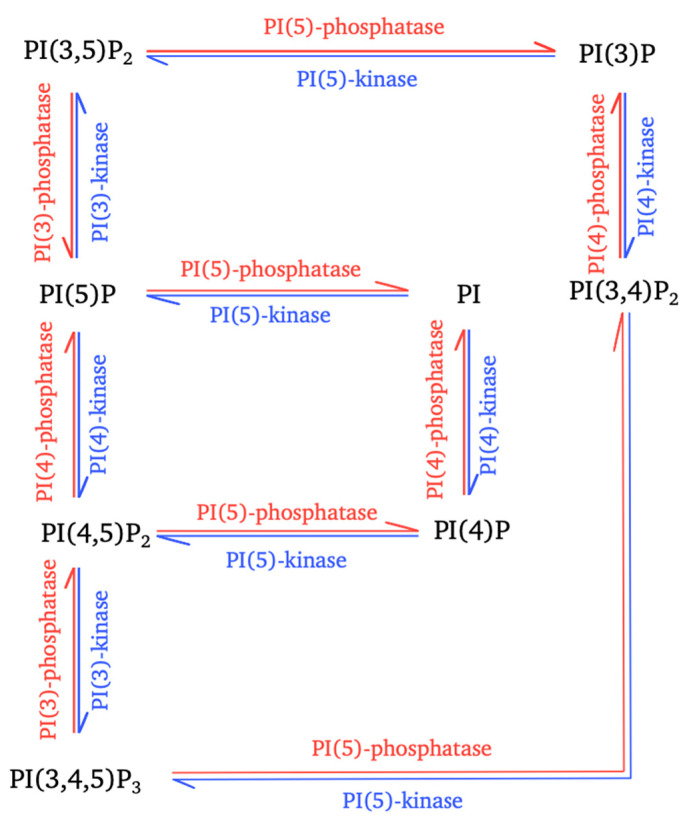
Interconversion of phosphoinositides mediated by cellular enzymes. The interconversion reactions for the controlled production of the seven types of phosphoinositides (PPIns) are shown. These reactions involve different phosphatidylinositol kinases (blue arrows) and phosphatidylinositol phosphatases (red arrows).

**Figure 2 viruses-12-01124-f002:**
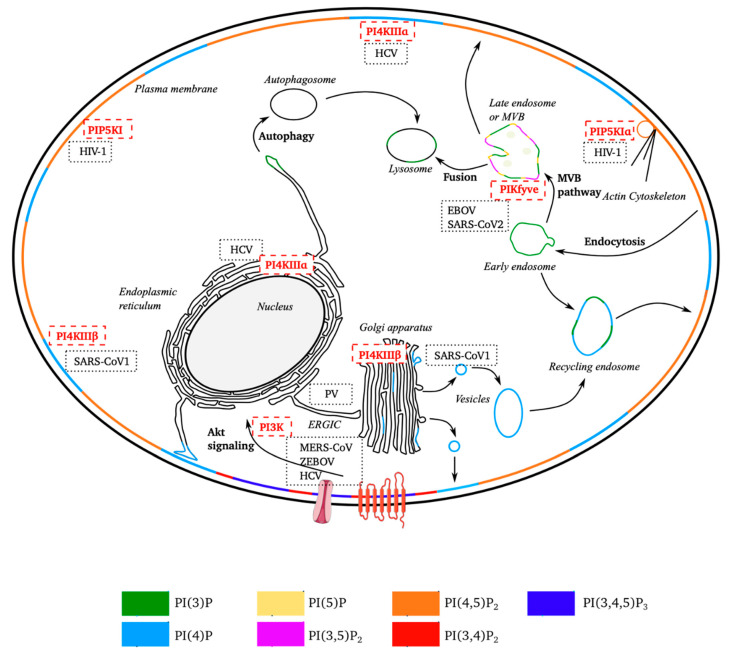
Schematic diagram of the subcellular distribution of phosphoinositides and sites targeted by positive-strand RNA viruses. Non-exhaustive representation of the subcellular distribution of the seven forms of phosphoinositides. Several cellular processes in which PPIns are also used are shown. The membranes of subcellular compartments are stained according to the types of PPIns predominantly present. The location of the kinases is indicated by their position, as are the sites targeted by the different RNA (+) viruses. (EBOV, Ebola virus; PV, poliovirus; SARS-CoV (1 and 2), severe acute respiratory syndrome coronavirus; MERS-CoV, Middle East respiratory syndrome-related coronavirus; HIV-1, human immunodeficiency virus; HCV, hepatitis C virus).

**Table 1 viruses-12-01124-t001:** Relationships between viruses and the kinases important for their life cycles.

Viruses		Kinases	Roles	References
Coronavirus	SARS-CoV-1	PI4KIIIβ	Involved in SARS-CoV-1 cellular entry via the spike protein by ACE2 receptor Increases PI(4)P in membrane organelle to create lipid environment suitable for SARS-CoV-1	[68]
	MERS-CoV	PI3K (Akt/mTOR, ERK/MAPK)	Allows MERS-CoV infection and replication by cell proliferation and apoptosis regulation	[69,70]
	SARS-CoV-2	PIKfyve	Involved in endosome maturation facilitating SARS-CoV-2 infection	[72]
Ebolavirus	EBOV	PIKfyve	Endosome maturation for EBOV entry by transport to NPC1-positive late endosomes VP40 interact with PI(3)P, PI(4,5)P_2_, PI(3,4,5)P_3_ to assembly and budding of viral particle	[73] [77,78]
	ZEBOV	PI3K	In relation to Akt, involved in viral entry	[75]
HCV		PI4KIIIα	Produces PI(4)P pool in the ER membrane for HCV replication (formation of membranous web structure) NS5A increase PI(4)P level by enhancing PI4KIIIα activity Indirectly involved in HCV secretion by PI(4)P production (interaction with GOLPH3 and MYO18A)	[80,81,82,83] [68,80] [80,86]
		PI3K	Allows Akt recruitment by interaction with NS5A, for cell survival and apoptosis regulation	[80]
Enterovirus (PV)		PI4KIIIβ	Coprecipitates with the viral 3A protein (formation of this complex during replication) for PI(4)P production Produces PI(4)P at the replication membrane (PI(4)P lipid-enriched organelle)	[82] [87]
HIV-1		PI5PKI	PI(4,5)P_2_ membrane production for HIV-1 assembly at the PM in T cells (knockdown induces Pr55Gag relocation)	[95,97,98,106]
		PIP5KIα	Actin cytoskeleton remodeling for HIV-1 entry in T cells By clustering, gp120 induces PI(4,5)P_2_ production by PIP5KIAα during viral contact with the cell (macrophages or T cells)	[88,89] [92]

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
