# Peer review of "The Role of Phosphatidylinositol Phosphate Kinases during Viral Infection"

_viruses, 2020, doi:10.3390/v12101124_

Round 1
Reviewer 1 Report
The review is summarizing interesting observations and findings of virus interaction with host phosphoinositide kinases. It is generally well organized and informative to the readers. However, errors are found, and clarification and modifications are suggested below to increase the quality of the review article.
1. Figure 2: It would be better to elaborate the name of kinases and related viruses in this figure. For example, In the table 1, PIKfyve is listed to regulate SARS-CoV-2 infection. However, in this figure, they do not come in close proximity at all. Additionally, colors are not distinguishable between PI(3,4,5)P3 and PI(3,5)P2 in the Figure.
2. Table 1: This table would be very informative for the readers, if polished/expanded. A few suggestions/corrections to make are listed as follows.
a) The references made in the table are not matching with those references found in the text. Please correct them.
b) Line 283, Table 1 was cited for VP40, but no description about VP40 of EBOV is found in the Table 1. Similarly, more viral proteins and cellular factors involved in the mechanism can be included in this table.
c) In case of HIV, as the authors discussed the difference between T cells and macrophages in the text, it would be better to mention if the description in the table is for T cells or macrophages or both in the table.
3. Do these kinases always “increase” virus replication? Perhaps it would be good to briefly discuss this in the text.
4. In many places, the names of phosphoinositide or enzymes with symbols are not clearly depicted, requiring proofreading them: please check the names out listed on the line numbers 105, 111, 117, 132, 142, 239, 393, etc.
Author Response
The review is summarizing interesting observations and findings of virus interaction with host phosphoinositide kinases. It is generally well organized and informative to the readers. However, errors are found, and clarification and modifications are suggested below to increase the quality of the review article
- Figure 2: It would be better to elaborate the name of kinases and related viruses in this figure. For example, In the table 1, PIKfyve is listed to regulate SARS-CoV-2 infection. However, in this figure, they do not come in close proximity at all. Additionally, colors are not distinguishable between PI(3,4,5)P3 and PI(3,5)P2 in the Figure.
Responses: We agree with the reviewer and we have modified the Figure 2:
- We have changed the color of the PI(3,5)P2 and now it appears in purple.
- We have introduced the viruses linked with the PI-Kinase mentioned in the text.
- Table 1: This table would be very informative for the readers, if polished/expanded. A few suggestions/corrections to make are listed as follows.
- a) The references made in the table are not matching with those references found in the text. Please correct them.
Response: The references have been corrected.
- b) Line 283, Table 1 was cited for VP40, but no description about VP40 of EBOV is found in the Table 1. Similarly, more viral proteins and cellular factors involved in the mechanism can be included in this table.
Response: The missing references were added in the Table 1.
- c) In case of HIV, as the authors discussed the difference between T cells and macrophages in the text, it would be better to mention if the description in the table is for T cells or macrophages or both in the table.
Response: We agree with the reviewer comment concerning the table 1 and we have specified the cell model used in case of HIV-1.
- Do these kinases always “increase” virus replication? Perhaps it would be good to briefly discuss this in the text.
Responses: In response to the reviewer’s request, the following sentence has been added in the conclusion of the revised manuscript (lines 414-416): “Regarding HCV and enterovirus, the increase in PI4K activity creates a localized pool of PPIns favoring their replication or their assembly.”
- In many places, the names of phosphoinositide or enzymes with symbols are not clearly depicted, requiring proofreading them: please check the names out listed on the line numbers 105, 111, 117, 132, 142, 239, 393, etc.
Response: We have checked and corrected all symbols
Reviewer 2 Report
In this manuscript, Beziau et al. describe the review of the role of phosphatidylinositol phosphate kinases (PIPKs) during viral infection. There is a number of reports showing the viral hijacking of cellular PIPKs during infection, so it is important to understand the mechanisms by which viruses exploit PIPKs. The authors provide a concise summary of the basic functions of PIPKs and their relationship to individual viruses.
Overall, I find that this review will be a useful introduction to the topic. However, the authors should correct or clarify the following.
- There are many typos. For example, Greek letters in protein names are broken in many places (see lines 105 and 111); note that there are more examples throughout the manuscript.
- Line 104; what does “C” stand for in PI3KCalph?
- Line 45, 51, and 52; PPIs should be changed to PPIns.
- Figure 2: I suggest that the authors use more distinctive colors for PI(3,4,5)P3 and PI(3,5)P2. They are difficult to distinguish in the figure.
- It is not clear how the authors selected the viruses in the manuscript. I suggest that the authors clarify what criteria they used to select the viruses in the introduction of section 4 (Kinases and viruses).
Author Response
In this manuscript, Beziau et al. describe the review of the role of phosphatidylinositol phosphate kinases (PIPKs) during viral infection. There is a number of reports showing the viral hijacking of cellular PIPKs during infection, so it is important to understand the mechanisms by which viruses exploit PIPKs. The authors provide a concise summary of the basic functions of PIPKs and their relationship to individual viruses.
Overall, I find that this review will be a useful introduction to the topic. However, the authors should correct or clarify the following.
- There are many typos. For example, Greek letters in protein names are broken in many places (see lines 105 and 111); note that there are more examples throughout the manuscript.
Response: The manuscript has been thoroughly checked and Greek letters are corrected.
- Line 104; what does “C” stand for in PI3KCalph?
Response: PI3KC is the other name of the PI3K type I. In the revised manuscript, we have added line 105: “(also named PI3KC)” after PI3K type I.
- Line 45, 51, and 52; PPIs should be changed to PPIns.
Responses: Changes have been made.
- Figure 2: I suggest that the authors use more distinctive colors for PI(3,4,5)P3 and PI(3,5)P2. They are difficult to distinguish in the figure.
Response: We have changed the color of the PI(3,5)P2, which now appears in purple in the modified manuscript
- It is not clear how the authors selected the viruses in the manuscript. I suggest that the authors clarify what criteria they used to select the viruses in the introduction of section 4 (Kinases and viruses).
Responses: In response to the reviewer’s request, the following sentence has been added in the section 4 of the revised manuscript (lines 231-233): “We have therefore chosen virus examples for which replication or assembly sites have a close relationship with membrane domains where major types of PI kinases are located.”